# Magnetic Guiding with Permanent Magnets: Concept, Realization and Applications to Nanoparticles and Cells

**DOI:** 10.3390/cells10102708

**Published:** 2021-10-09

**Authors:** Peter Blümler

**Affiliations:** Institute of Physics, University of Mainz, Staudingerweg 7, 55099 Mainz, Germany; bluemler@uni-mainz.de; Tel.: +49-6131-392-4240

**Keywords:** steering, magnetic force, magnetic drug targeting (MDT), nanoparticle, SPIO, ferrofluid, superparamagnetic, ferromagnetic, Halbach magnets, dipole, quadrupole, cells, micro-robots, endoscopic capsules, magnetic resonance imaging, MRI, magnetic particle imaging, MPI

## Abstract

The idea of remote magnetic guiding is developed from the underlying physics of a concept that allows for bijective force generation over the inner volume of magnet systems. This concept can equally be implemented by electro- or permanent magnets. Here, permanent magnets are in the focus because they offer many advantages. The equations of magnetic fields and forces as well as velocities are derived in detail and physical limits are discussed. The special hydrodynamics of nanoparticle dispersions under these circumstances is reviewed and related to technical constraints. The possibility of 3D guiding and magnetic imaging techniques are discussed. Finally, the first results in guiding macroscopic objects, superparamagnetic nanoparticles, and cells with incorporated nanoparticles are presented. The constructed magnet systems allow for orientation, movement, and acceleration of magnetic objects and, in principle, can be scaled up to human size.

## 1. Introduction

In this review, the meaning of *magnetic guidance* is understood as a remote, untethered and contact-free control of the movements of an object via magnetic interactions. The movements should happen on arbitrary trajectories inside a container caused by an external device.

Typical examples of such magnetically guided objects are endoscopic capsules for inspection of the gastrointestinal tract or superparamagnetic nanoparticles suggested for local therapy, which therefore have to be moved through blood vessels. There are numerous reviews on the subject because this research area is very diverse and the problem has been tackled from different directions. The following reviews on magnetically guided medical devices [1,2,3]; miniature robots [4]; nanoparticles in microfluidics and nanomechanics [5] for drug delivery [6,7,8], hyperthermia, and alternative local magnetic therapeutic effects [9,10]; tissue engineering [11,12,13]; as well as magnet systems for this purpose [14] are some of the most recent (or a book [15] treating most of these topics). There are many more applications of magnets in biomedicine, e.g., permanent magnets are also used for the separation of superparamagnetic nanoparticles from solution. If the nanoparticles are functionalized, specific substances can be removed from the solution. However, this is not guiding in the sense of the initial definition because the direction of motion of the particles is not controllable.

It is not the intention of this review to summarize this very active, vast, and diverse field of research, but rather to discuss a simple and very general concept of magnetic guiding that borrows ideas from the treatment of magnetic fields in magnetic resonance imaging (MRI). To avoid confusion, this comparison does not imply that the magnetic fields in MRI are particularly useful for guiding (although possible [16]), because the typical gradient fields are too weak. It is rather the physical principle that is similar. Both techniques use a projection of a small deflection or encoding magnetic field tensor on a much stronger and homogeneous field (component), which generates spatial bijection. In the case of MRI, the unavoidable concomitant gradient components of the coil systems can be ignored if the strength of the homogeneous field is much higher (a concept which fails at low magnetic fields). This physical statement and how magnetic guiding of paramagnetic objects can be implemented analogously will be explained in the next section. From this perspective, it does not matter if the magnetic fields for guiding are generated by electro-, permanent, or hybrid magnets. However, since permanent magnets have so many advantages over electromagnets, only realizations using exclusively permanent magnets are discussed in Section 3. The described bijective guiding concept relies on the experimental condition that the homogeneous field is much stronger than the deflection field, and Section 4 describes what happens if this is violated. While guiding macroscopic objects does not require particularly strong magnetic fields, doing alike with superparamagnetic nanoparticles is a challenge. It is almost unavoidable that their mutual magnetic interactions induce cluster formation. This changes not only the magnetic force but also the hydrodynamics as discussed in Section 5. Section 6 then covers possible 3D realizations of such guiding instruments and magnetic imaging schemes. Finally, some first applications of the technique are shown before this article concludes. These first applications demonstrate that the proposed concepts work nicely in two dimensions by guiding macroscopic objects as well as suspended superparamagnetic nanoparticles or cells with incorporated nanoparticles.

At this point, a few general remarks need to be made. The following line of arguments will consider paramagnetic materials only, i.e., materials with a relative permeability bigger than one (μ_r_ > 1) or positive magnetic susceptibilities (χ = μ_r_ − 1), hence including ferri-, ferro-, and superparamagnetic materials. In principle, diamagnetic (0 ≤ μ_r_ < 1) substances can be guided with the same instrument as well. Essentially, it would just reverse the sign in the force equations. However, since the only substances with a diamagnetic permeability significantly different from 1 are superconductors (with μ_r_ = 0) and most applications of magnetic guiding are typically in biological systems, they will be ignored in the following.

Since most of this article deals with concepts, the problem is that most of the time these concepts are idealized to the task of guiding a magnetic dipole with a magnetic moment $\vec{m}\;[{\mathrm{Am}}^{2}]$ by an applied magnetic flux density $\vec{B}\;[T]$. Deviations when concerned with real bulk materials are discussed in Appendix A. At this point, the author wants to apologize for not always using the completely correct terminology concerning magnetic flux density, *B* [T], which in the following is often termed as magnetic field or in-duction field to improved readability. Wherever a physical magnetic field, *H* [A/m], is meant, it is labelled as such. For most of this conceptual presentation, *B* = μ_0_
*H* is valid with μ_0_ ≈ 4π × 10^−7^ N/A^2^ is the vacuum permeability.

To further improve readability, highly technical or mathematical details were separated in seven appendices, which are found at the end of this article. They might be particularly useful for readers who want to design their own systems or follow the derivation of equations.

The author wants to conclude this introduction with a statement about names. The first system of this kind was nicknamed “MagGuider” (for **Mag**netic **Guid**ing and Scann**er**) [17], but this name should not be used here, because this article rather deals with a concept than with a particular instrument.

## 2. Concept of Magnetic Guiding

Magnetic guiding has been an established technique since 1897 [18], when Ferdinand Braun invented magnetic guidance of charged particles (electrons or ions) by cathode ray tubes where the electrons are emitted from a cathode into an evacuated tube, accelerated by an anode, and deflected by magnetic fields (used en masse in analogue oscilloscopes and television screens). The magnetic deflection is based on the Lorentz force ${\vec{F}}_{L}=q\;\vec{v}\times \vec{B},$ which is perpendicular to the direction of the magnetic flux density $\vec{B}$ and the flight direction of the particles with charge *q* and velocity $\vec{v}$. However, the situation is very different if an electrically neutral paramagnetic material is exposed to magnetic fields. The force is then the gradient $\left(\vec{\nabla }=\left[\partial /\partial x,\;\partial /\partial y,\;\partial /\partial z\;\right]\right)$ of the magnetic field acting on the object with a magnetic moment $\vec{m}$
(1)$${\vec{F}}_{m}=\vec{\nabla }\left(\vec{m}\cdot \vec{B}\right)\;\overset{\mathrm{Appendix}\;A}{≃}\;\left(\vec{m}\cdot \vec{\nabla }\right)\vec{B}.$$

The right simplified term is usually correct for the applications discussed here, however, it is not generally the case. Particularly, it assumes that *m* is not dependent on *B*, which depends on the material and the range of *B* (see Appendix A for a discussion).

So what happens to a small paramagnetic object in an inhomogeneous magnetic field? It is hard to imagine that an object that should be guided through space is not freely movable (at least in two dimensions). If the object has an intrinsic fixed direction of $\vec{m}$ (e.g., remanent magnetization), it is rotated by the magnetic torque
(2)$${\vec{\tau }}_{m}=\vec{m}\times \vec{B}.$$
until the cross-product becomes zero or $\vec{m}$ is parallel to $\vec{B}$. If the object has initially (at *B* = 0) no preferred direction of $\vec{m}$, the actual field will magnetize it (orient the electron spins) along $\vec{B}$. Either way, as a result, $\vec{m}$ points along $\vec{B}$, which is very unfortunate with respect to guiding, because the dot-product in Equation (1) will lose its sign for two parallel vectors and the material will always move towards higher magnetic fields (cf. Figure 1a,b). This is an everyday observation, as e.g., paper clips are attracted equally by the north and south pole of a permanent magnet. For steering this is like using a clipper without a sail. Almost independently of what one tries with the rudder, the boat will go to where the winds or currents move it. In electrodynamics, this is also known as Earnshaw’s theorem [19], and it is the reason why permanent magnets were originally not considered as being useful for magnetic guidance, because as their name suggests they are permanent and cannot be switched on or off.

Now the question arises why guiding charged particles is so straightforward, while it is so difficult to control the collective spin of electrons in materials magnetically? The reason is the bijective direction ($\vec{v}$) of the electron beam, which is just slightly deflected by steering fields. This suggests that a preferred direction would also be beneficial for steering paramagnetic objects. This is tantamount to a magnetic field that just orients (polarizes) the particles without exerting a force on them. For static magnetic fields, this request can be fulfilled by applying a strong but homogeneous magnetic flux density, *B*_hom_, which magnetizes the object along its direction. An additional, small, and spatially-dependent steering or deflecting field can then act as a perturbation but with full directional control (cf. Figure 1c,d). Ideally, this deflecting field will have a linear spatial dependence, i.e., a constant gradient (The fact that *G* is a tensor is ignored for the moment), $\vec{\nabla }\vec{B}=G$, and the total field in such an experiment is then
(3)$$\vec{B}(r)={\vec{B}}_{\mathrm{hom}}+\;G\vec{r}.$$

With the reasonable assumption that there is no strong spatial variation of the magnetic moment over the sample, one could conclude that *F*_m_ = *mG* (because $\vec{\nabla }{\vec{B}}_{\mathrm{hom}}=0$). Under certain limits this is correct, but unfortunately magnetism is not quite that simple. Things become a bit more complicated due to Maxwell’s (or Gauss’) law
(4)$$\vec{\nabla }\cdot \vec{B}=\frac{\partial B_{x}}{\partial x}+\frac{\partial B_{y}}{\partial y}+\frac{\partial B_{z}}{\partial z}=0.$$

Hence, there cannot be a single gradient field at any point. Either the field has to be homogeneous or the sum of all its spatial derivatives have to cancel. For the simple case of a perfect quadrupolar field (see Appendix B), this could be for instance $\partial B_{x}/\partial x=+G$ and $\partial B_{y}/\partial y=-G$, consequently Equation (4) dictates $\partial B_{z}/\partial z=0$. Then a more detailed representation of Equation (3) will be
(5)$$\begin{array}{ll} \vec{B}(\vec{r})\; & =\left[\begin{array}{c} B_{x}(x,y,z)\\ B_{y}(x,y,z)\\ B_{z}(x,y,z) \end{array}\right]={\vec{B}}_{\mathrm{hom}}+\bar{\bar{G}}\vec{r}\\ & =B_{\mathrm{hom}}\left[\begin{array}{c} 1\\ 0\\ 0 \end{array}\right]\;+\;G\left[\begin{array}{ccc} 1 & 0 & 0\\ 0 & -1 & 0\\ 0 & 0 & 0 \end{array}\right]\left[\begin{array}{c} x\\ y\\ z \end{array}\right]=\left[\begin{array}{c} B_{\mathrm{hom}}+Gx\\ -Gy\\ 0 \end{array}\right]. \end{array}$$

The deflecting field is written here in the most general form as a gradient tensor ($\bar{\bar{G}}$, see Appendix F), which will be needed later. As discussed above, the magnetic moment of an object at $\vec{r}={\left[x,y,z\right]}^{T}$ will be oriented parallel to $\vec{B}(\vec{r})$ (with unit vector ${\hat{e}}_{B}$)
(6)$$\begin{array}{cc} \vec{m}(x,y,z) & =\left|\vec{m}\right|\;{\hat{e}}_{B}\;=\;\left|\vec{m}\right|\;\frac{\vec{B}}{\left|\vec{B}\right|}\;\\ & =\frac{\left|\vec{m}\right|}{\sqrt{{\left(B_{\mathrm{hom}}+Gx\right)}^{2}+G^{2}y^{2}}}\left[\begin{array}{c} B_{\mathrm{hom}}+Gx\\ -Gy\\ 0 \end{array}\right]\;\overset{B_{\mathrm{hom}}\gg Gr}{\approx }\;\left|\vec{m}\right|\;\left[\begin{array}{c} 1\\ 0\\ 0 \end{array}\right]\;. \end{array}$$

The last approximation was already motivated in the discussion of Figure 1 and is the origin of bijection, namely that the homogeneous field must be much stronger than the local deflection field, so that its tensorial properties can be reduced to a vector via projection. The condition for this prerequisite is then
(7)$$\left|{\vec{B}}_{\mathrm{hom}}\right|\;\gg \;\left|\vec{\nabla }\vec{B}\right|\left|\vec{r}\right|.$$

A full treatment will follow but to clarify the concept, it is instructive to continue with the approximation from Equation (6). Then the magnetic force in Equation (1) simplifies to
(8)$$\begin{array}{cc} {\vec{F}}_{m}(x,y,z) & =\;(\vec{m}\cdot \vec{\nabla }\;)\vec{B}\approx \;\left|\vec{m}\right|\;\frac{\partial }{\partial x}\;\left[\begin{array}{c} B_{\mathrm{hom}}+Gx\\ -Gy\\ 0 \end{array}\right]\\ & =\;\left|\vec{m}\right|G\;{\hat{e}}_{x}\;, \end{array}$$
or more generally only the field component of the deflection field, which is parallel to ${\vec{B}}_{\mathrm{hom}},$ determines the direction and amplitude of the magnetic force. It is a very beneficial feature of this concept that there is no spatial dependence of the force vector in Equation (8), hence the guiding force is homogeneous or constant over that region where Equation (7) is fulfilled (cf. also Figure 5). This is an important issue because other systems which guide an object by moving permanent magnets around the outside of the container (e.g., [20]) or use electromagnets on opposing ends of the container, also have to consider the non-linear drop of the magnetic field with distance (depending on their dimensions, the far-field of permanent magnets drops with an exponent between −2 and −3, and hence the force with −3 to −4). This can extremely complicate the control because the position of the object has to be known precisely to estimate speed and direction of motion. This is a problem that does not exist in the presented concept. Additionally, the movements are also not limited to the direction of $\vec{B}$. In the following section and Appendix C it is explained how any direction can be addressed by rotating the gradient field relative to ${\vec{B}}_{\mathrm{hom}}.$

## 3. Permanent Magnets with Adjustable Fields

As already said in the introduction, this conceptual idea of magnetic guiding can of course be implemented with any magnet system consisting of either electro-, permanent magnets, or hybrids. While Equation (8) motivates this concept, real guiding of objects along arbitrary paths needs deflection fields, which must become time-dependent in orientation and amplitude [21]. Since permanent magnets are not typically associated with these properties, they are often excluded from considerations, although they offer some important advantages over electromagnets. Particularly, if the devices need to be scaled up, the enormous power consumption of resistive electromagnets becomes a real problem. For example, to generate roughly the magnetic field produced by 1 cm^3^ of modern rare-earth magnets, several kW of electrical power are already needed and additionally the generated heat must be removed by cooling. This can be estimated from the Amperian loop model, where a loop of current, *I*, produces a magnetic moment *m* = *IA*, with *A* as the surface of the loop. Using Equation (A3) (see Appendix A), this can be rearranged for a cylinder of height *h* to *I* = *B*_R_
*h*/μ_0_. Hence, a cylinder of rare earth material with a remanence *B*_R_ = 1.3 T and *h* = 1 cm height is equivalent to a current of ca. 10 kA. Additionally, the time-dependence of large coils is very nonlinear, as the inductance scales with the square of the number of windings and the coil cross-section. Large inductances then result in long delays for discharging and charging the coil. Together with the temperature and hence resistance changes associated with time-dependent currents, the resulting magnetic fields are difficult to calculate and control [22,23]. Superconducting magnets are not an acceptable option either because they cannot be switched fast. So what about permanent magnets? As the name suggests, the orientation and strength of a permanent magnet are time-independent, however, many of them can be arranged to systems such that the direction and strength of their resulting magnetic fields can be changed by simple mechanical rotation [24].

The most suitable arrangements for this purpose are so-called Halbach cylinders [25]. Their concept, construction, and field calculation is discussed in Appendix B. In order to implement the concept of magnetic guidance from Section 2, the homogeneous field will be generated by a Halbach (inner) dipole (see Figure 2a), while the constant gradients can readily be provided by a Halbach (inner) quadrupole (see Figure 2b). In the following, the discussion will be limited to ideal systems. Hence, it is sufficient to treat this as a two-dimensional problem (infinite length in the third dimension, see Appendix B).

The first advantage of such Halbach cylinders is that they provide ideal homogeneous and graded fields (as are assumed for Equation (8)) with simple geometric relations to calculate their field
(9)$$\begin{array}{cc} \mathrm{dipole}: & \;\vec{B}(x,y)=B_{R}\ln\frac{R_{o}}{R_{i}}\;\left[\begin{array}{c} 1\\ 0 \end{array}\right]=B_{D}\left[\begin{array}{c} 1\\ 0 \end{array}\right]\;,\\ \mathrm{quadrupole}: & \;\vec{B}(x,y)\;=\;2B_{R}\left(\frac{1}{R_{i}}-\frac{1}{R_{o}}\right)\;\left[\begin{array}{cc} 1 & 0\\ 0 & -1 \end{array}\right]\;\left[\begin{array}{c} x\\ y \end{array}\right]=G_{Q}\;\left[\begin{array}{cc} 1 & 0\\ 0 & -1 \end{array}\right]\left[\begin{array}{c} x\\ y \end{array}\right]\;, \end{array}$$
where *R*_i_ is the inner and *R*_o_ the outer radius of the hollow cylinders, and *B*_R_ [T] is the remanence of the used permanent magnet material. The strength of the homogeneous field, *B*_D_ = *B*_hom_, of the dipole and the strength of the gradient, *G*_Q_, produced by the quadrupole, are now indexed by the type of Halbach magnet they originate from. This will help to retain an overview when nesting multiple rings.

The second great advantage is the absence of stray fields, so that they are “no magnets” when approached from the outside. Therefore, the cylinders can be concentrically arranged or nested and mutually rotated without much torque [24]. If two Halbach cylinders of the same type are nested and the geometries are chosen such that they both produce the same field or gradient strength, their combined field can then be varied between zero and twice the value of a single cylinder. This allows to scale the field or force or eventually even switch it off. This principle is illustrated in Figure 3.

If the example from Equation (5) is put into effect by a Halbach dipole and a Halbach quadrupole, and the quadrupole is rotated by an angle α relative to the dipole, the magnetic field in such a structure is (cf. Appendix C and Figure 3l)
(10)$$\begin{array}{cc} \vec{B}(x,y) & =B_{D}\left[\begin{array}{c} 1\\ 0 \end{array}\right]+G_{Q}\;\left[\begin{array}{cc} \cos2\alpha & \sin2\alpha \\ \sin2\alpha & -\cos2\alpha \end{array}\right]\left[\begin{array}{c} x\\ y \end{array}\right]\\ & =\left[\begin{array}{c} B_{D}+G_{Q}\left(x\cos2\alpha +y\sin2\alpha \right)\\ G_{Q}\left(x\sin2\alpha -y\cos2\alpha \right) \end{array}\right]. \end{array}$$

By using the arguments of Equations (5) and (6), the field component that is not along *B*_D_ (i.e., *B_y_*) can be ignored and the magnetic force is given by
(11)$$\begin{array}{cc} {\vec{F}}_{m} & =\vec{\nabla }(\vec{m}\cdot \vec{B})\approx \vec{\nabla }\left|\vec{m}\right|B_{x}\\ & =\left|\vec{m}\right|\left[\begin{array}{c} \partial /\partial x\\ \partial /\partial y \end{array}\right]\left(B_{D}+G_{Q}\left(x\cos2\alpha +y\sin2\alpha \right)\right)=\left|\vec{m}\right|G_{Q}\left[\begin{array}{c} \cos2\alpha \\ \sin2\alpha \end{array}\right]. \end{array}$$

This means that the force has a constant strength of $\left|\vec{m}\right|G_{Q}$ and rotates with 2α over the entire volume where the prerequisite of Equation (7) is fulfilled. Although it is a bit counterintuitive that the object moves at twice the angle of which its actuator is rotated, this concept gives complete control over the direction of a magnetically guided object in such a magnet system [17].

In order to completely control the movements of such a guided object, not only the direction but also the amplitude of the force must be controlled. This can easily be done by using a second quadrupole, ideally of a size that produces the same gradient strength in the internal volume as already provided by the first quadrupole (cf. Appendix B). The direction of the force shall be determined by α and shall not be altered by scaling the force; one quadrupole must be rotated by an angle (α + β/2) and the other by (α − β/2)
(12)$$\begin{array}{cc} \vec{B} & =\;\left[\begin{array}{c} B_{D}+G_{Q}\left(x\cos\left(2(\alpha +\frac{\beta }{2})\right)+y\sin\left(2(\alpha +\frac{\beta }{2})\right)\right)+G_{Q}\left(x\cos\left(2(\alpha -\frac{\beta }{2})\right)+y\sin\left(2(\alpha -\frac{\beta }{2})\right)\right)\\ G_{Q}\left(x\sin\left(2(\alpha +\frac{\beta }{2})\right)-y\cos\left(2(\alpha +\frac{\beta }{2})\right)\right)+G_{Q}\left(x\sin\left(2(\alpha -\frac{\beta }{2})\right)-y\cos\left(2(\alpha -\frac{\beta }{2})\right)\right) \end{array}\right]\\ & =\;\left[\begin{array}{c} B_{D}+G_{Q}\left\{x\left(\cos\left(2\alpha +\beta \right)+\cos\left(2\alpha -\beta \right)\right)+y\left(\sin\left(2\alpha +\beta \right)+\sin\left(2\alpha -\beta \right)\right)\right\}\\ G_{Q}\left\{x\left(\sin\left(2\alpha +\beta \right)+\sin\left(2\alpha -\beta \right)\right)-y\left(\cos\left(2\alpha +\beta \right)+\cos\left(2\alpha -\beta \right)\right)\right\} \end{array}\right]\\ & =\;\left[\begin{array}{c} B_{D}+G_{Q}\left\{x\left(2\cos\beta \cos\left(2\alpha \right)\right)+y\left(2\cos\beta \sin\left(2\alpha \right)\right)\right\}\\ G_{Q}\left\{x\left(2\cos\beta \sin\left(2\alpha \right)\right)-y\left(2\cos\beta \cos\left(2\alpha \right)\right)\right\} \end{array}\right]\\ & =\;\left[\begin{array}{c} B_{D}+2\cos\beta \;G_{Q}\left\{x\cos\left(2\alpha \right)+y\sin\left(2\alpha \right)\right\}\\ 2\cos\beta \;G_{Q}\left\{x\sin\left(2\alpha \right)-y\cos\left(2\alpha \right)\right\} \end{array}\right]\;. \end{array}$$ This generates a force (again only taking the direction of *B*_D_ into account)
(13)$$\begin{array}{cc} {\vec{F}}_{m} & \approx \;\left|\vec{m}\right|\;\vec{\nabla }B_{x}\\ & =\;\left|\vec{m}\right|\;\left[\begin{array}{c} \partial /\partial x\\ \partial /\partial y \end{array}\right]\;\left(B_{D}+2\cos\beta \;G_{Q}\left(x\cos2\alpha +y\sin2\alpha \right)\right)=2\cos\beta \;\left|\vec{m}\right|\;G_{Q}\left[\begin{array}{c} \cos2\alpha \\ \sin2\alpha \end{array}\right]\;. \end{array}$$

From Figure 3l it can clearly be seen that the angle β between the quadrupoles only scales the force by 2$\left|\cos\beta \right|$, i.e., from twice the gradient generated by one quadrupole at β = 0° and 180° and zero at β = 90° and 270°. At the same time, the direction of the force is kept constant at 2α. In this way, it is possible to accelerate, decelerate, stop, or move the object at very sharp angles. This is best illustrated by a video in which a small steel ball (in very a highly viscous medium to slow down its speed) was used in such a system to write letters (Figure 9a).

A complete guiding system that allows not only changing the direction and strength of the magnetic force but also the direction and strength of *B*_hom_ will then consist of two dipoles and two quadrupoles. With such an instrument, all magnetic fields can then be cancelled in the inner volume, so that also aggregates of nanoparticles might disintegrate (see Section 5). Such a system is treated in a very general way in Appendix D. The resulting equations may look complicated at first glance, however, they contain only simple trigonometric functions that can easily be calculated. This is a fact that should not be underestimated, because there are neither time-dependent nor interdependent terms, nor non-linearities in these equations. Even if the magnetic fields cannot be made as ideal as assumed here, they can be measured and tabulated for each ring and used for the calculation of the local force with high precision.

There is no torque if nested ideal Halbach systems are rotated with respect to each other [26]. However, the necessary segmentation and truncation to a certain length (see Appendix B) also introduces a torque ${\vec{\tau }}_{m}\;∼\;\sin(k\alpha )$ [27] whose amplitude very much depends on the geometry and has its main contributions due to cogging at the edges of the segments. Especially for quadrupoles constructed from polygonal magnets, the torque can become significant.

It may be useful to evaluate the range or dimensions of magnetic fields, gradients and instruments sizes. One limit is the demagnetization field, which limits the local magnetic fields inside the magnet structure. If exceeded, the magnetic material will alter its magnetization like in the initial polarization process (to some extent, this is discussed in Appendix B and in [28,29]). However, this process is ignored and a simple guiding system made from one Halbach quadrupole (*R*_i_ = *r*_1_, *R*_o_ = *r*_2_) surrounded by a Halbach dipole (*R*_i_ = *r*_2_, *R*_o_ = *r*_3_) is straightforwardly calculated from Equation (9). Then the field strength, *B*_D_, of an ideal Halbach-dipole and the gradient strength, *G*_Q_, of a quadrupole are given by Equation (9) and combined with Equation (7)
(14)$$\begin{array}{l} G_{Q}=2B_{R}\;\left(\frac{1}{r_{1}}-\frac{1}{r_{2}}\right)\;,\\ B_{D}=B_{R}\;\ln\frac{r_{3}}{r_{2}}\;,\\ B_{D}\;\geq \;G_{Q}\;r_{1}\;. \end{array}$$

From this set of equations, the following relations can be equated:(15)$$r_{3}\geq r_{2}\;\exp\left(2\;\frac{r_{2}-r_{1}}{r_{2}}\right)\;\mathrm{and}\;B_{D\;}=2B_{R}\left(1-\frac{r_{1}}{r_{2}}\right)\;\mathrm{and}\;G_{Q}\leq \frac{B_{D}}{r_{1}}$$

The dependency of this equation and the resulting homogeneous and graded fields are shown in Figure 4 assuming *r*_1_ and *B*_R_ are given and the relation in Equation (15) is treated as an equation. Although using the equations for ideal Halbach-magnets in Equation (14) is somewhat naïve (see Appendix B), Figure 4 gives a good estimate for the magnitudes of the geometric dimensions, magnetic fields, and gradients, which are important to estimate the achievable magnetic forces on nanoparticles in Section 5.

## 4. Deviation from Constant Forces

The key requirement for obtaining a constant force over the internal volume of radius *R* is given by Equation (7). This defines a prerequisite for ignoring the unavoidable additional components of the deflection field. In magnetic resonance imaging (MRI), these components are also named concomitant gradients [30]. Figure 5 gives a visual explanation of what happens to the force field if the gradients become too strong. The deviations are of course strongest at large distances from the center and that central line, which is perpendicular to the anticipated force direction (vertical central line in Figure 5). This is because the other concomitant gradient field has a zero here as well (cf. Figure 2c,d).

Nevertheless, the guiding equation can still easily be solved by using Equation (6) without the approximation of Equation (7) (see also Appendix D for a full treatment). For instance, the magnetic field, $\vec{B}$, in Equation (10) for a dipole with rotated quadrupole gives then the following orientation for the local magnetic moment
(16)$$\begin{array}{cc} \vec{m}(x,y)= & \;\left|\vec{m}\right|\;\frac{\vec{B}}{\left|\vec{B}\right|}=\frac{\left|\vec{m}\right|}{\left|\vec{B}\right|}\;\left[\begin{array}{c} B_{D}+G_{Q}\left(x\cos2\alpha +y\sin2\alpha \right)\\ G_{Q}\left(x\sin2\alpha -y\cos2\alpha \right) \end{array}\right]\\ & \mathrm{with}\;\left|\vec{B}\right|=\sqrt{B_{D}^{2}\;+\;G_{Q}^{2}\left(x^{2}+y^{2}\right)\;+2B_{D}G_{Q}\left(x\cos2\alpha +y\sin2\alpha \right)}\;, \end{array}$$
and the force becomes
(17)$${\vec{F}}_{m}(x,y)=\frac{\left|\vec{m}\right|G_{Q}}{\left|\vec{B}\right|}\;\left[\begin{array}{c} G_{Q}x+B_{D}\cos2\alpha \\ G_{Q}y+B_{D}\sin2\alpha \end{array}\right].$$

The deviation between the ideal and real case can be described by a deviation angle (i.e., the angle between the magenta and blue arrows in Figure 5). Its maximum, δ_max_, is derived in [17] as
(18)$$\delta_{\max}=\sin^{-1}\left(\frac{GR}{B}\right)\;\mathrm{or}\;G=\frac{B}{R}\sin\delta_{\max}.$$

From this equation, it is obvious that absolute angular precision (δ_max_ = 0) implies *G* = 0 and hence no force or deflection, except for the line where the concomitant gradient is zero. For an angular error of δ_max_ ≤ 1°, a ratio *B*/(*GR*) ≈ 60 must be achieved. However, the situation is not as bad as it seems, because there are simple analytical expressions of this deviation for every spatial coordinate so that the error can easily be accounted for (see Appendix D) and usually the real challenge is to guide deep inside (center, or small *R*) the body where the precision is naturally higher.

## 5. Magnetic Force and Velocity

The magnetic force, ${\vec{F}}_{m}$, on a single particle in such magnet systems with a magnetic field $\vec{B}(\vec{r})$, which has a gradient $\vec{G}(\vec{r})$ at the spot$,\;\vec{r},$ of the particle, is given by Equation (1) or (8).
(19)$${\vec{F}}_{m}(\vec{r})=\left|\vec{m}\right|\vec{G}(\vec{r})\approx \left|\vec{M}(\vec{B}(\vec{r}))\right|\;V\;\vec{G}(\vec{r})\approx \left|{\vec{M}}_{s}\right|V\vec{G}(\vec{r}),$$
where the magnetic moment is expressed by the more commonly used magnetization, *M* [A/m], and the particle’s volume, *V.* However, the magnetization is the volume integral of all magnetic moments in the object, which do not necessarily all align with the external magnetic flux density, and consequently, $\vec{M}$ is a function of $\vec{B}(\vec{r})$. If all magnetic moments are oriented, the material is saturated with a magnetization, *M*_s_. This is assumed to happen for the last term in Equation (19) and is usually a valid assumption for most magnetic materials in nanoparticle synthesis (e.g., for magnetite, Fe_3_O_4_: $M_{s}^{m}\;:=M_{s}/\rho$ ≈ 4–80 Am^2^/kg with a density ρ = 1000–5200 kg/m^3^, depending on if the particle composition has a saturation induction field of *B* < 10 mT).

In the following it is further assumed that the particle has the shape of a sphere and is embedded in a medium of dynamic viscosity, η [Pa s]. Then the medium will exert a Stokes friction or drag of
(20)$${\vec{F}}_{S}=6\pi \eta R_{h}\vec{v},$$
on the particle, so that in equilibrium it will move with constant velocity $\vec{v}$.
(21)$$\vec{v}\;=\;\frac{\left|{\vec{M}}_{s}\right|V}{6\pi \eta R_{h}}\;\vec{G}\;=\;\frac{2\left|{\vec{M}}_{s}\right|R^{3}}{9\eta R_{h}}\;\vec{G}\;\overset{\mathrm{if}\;R=R_{h}}{=}\;\frac{2\left|{\vec{M}}_{s}\right|R^{2}}{9\eta }\;\vec{G}.$$

The hydrodynamic radius of the particle, *R*_h_, does not have to be identical to the geometric radius (as assumed for the last approximation) of the magnetic part(icle) of the spherical object, because many nanoparticles consist of a magnetite core and some biocompatible shell [31]. However, in most applications of magnetic guiding of nanoparticles, more than just one of such particles is administered. Therefore, it is very likely that neighboring particles will form chains, because they are all magnetized in the same direction and will experience dipolar interaction (cf. Figure 6a–d, this clustering can only be avoided with low concentrations or large *R*_h_ together with low *M*_s_ so that the interaction energy is lower than the thermal energy).

From the perspective of transportation, such clustering is advantageous, because a cluster of *n* particles experiences an up to *n*-fold increase in force [32]. Associated with the clustering is obviously a change of the shape of the guided object, and hence Equation (21) is no longer valid. If a cloud of superparamagnetic nanoparticles is injected inside a magnet system designed as suggested above, the strong and homogeneous magnetic field magnetizes all particles along its direction. If no gradient (=neither force nor velocity) is present, cluster-formation will only happen on the time-scale of Brownian motion (i.e., self-diffusion). Gradients will assist a quick cluster-formation as the particles are all moved in the same direction [33], and hence their inter-distances will become smaller in this direction. As soon as some larger clusters have formed, they will attract neighboring particles with increased force. A self-accelerating process starts in which the particles are mainly attracted to the ends of the forming beaded structure (cf. Figure 6b). This is exactly the same process that forms “field lines” from iron-filings in the stray field of magnets (cf. Figure 6b,c).

To account for this behavior, the shape of such a beaded chain is approximated by a long slender body of length *L*_h_ = 2*nR*_h_ (cf. Figure 6a) and its velocity is given by (Equations (7.10) and (7.21) in [34])
(22)$$\begin{array}{c} \vec{v}=\left[\begin{array}{c} v_{∥}\;\cos\alpha \\ v_{⊥}\;\sin\alpha \end{array}\right]\;\mathrm{with}\;\alpha \;∡\left(\vec{m},\;{\vec{F}}_{m}\right)\\ \mathrm{and}\;{\vec{F}}_{m}\;\approx \;n\;\left|{\vec{M}}_{s}\right|V\vec{G}=n\;\left|{\vec{M}}_{s}\right|\frac{4}{3}\pi R^{3}\vec{G}, \end{array}$$
(23)$$\begin{array}{cc} & v_{∥}\approx \left|{\vec{F}}_{m}\right|\frac{\ln\left(\frac{L_{h}}{R_{h}}\right)+C_{∥}}{2\pi \eta L_{h}}\approx \left|{\vec{F}}_{m}\right|\frac{\ln\left(2n\right)+C_{∥}}{4\pi \eta \;nR_{h}}\overset{\mathrm{if}\;R\;=\;R_{h}}{=}\;2\kappa \left[\ln(2n)+C_{∥}\right]\;,\\ \mathrm{with} & v_{⊥}\approx \left|{\vec{F}}_{m}\right|\frac{\ln\left(\frac{L_{h}}{R_{h}}\right)+C_{⊥}}{4\pi \eta L_{h}}\approx \left|{\vec{F}}_{m}\right|\frac{\ln\left(2n\right)+C_{⊥}}{8\pi \eta \;nR_{h}}\overset{\mathrm{if}\;R\;=\;R_{h}}{=}\kappa \left[\ln(2n)+C_{⊥}\right]\\ & \mathrm{with}\;\kappa \;=R^{2}\left|{\vec{M}}_{s}\right|\left|\vec{G}\right|/6\eta . \end{array}$$ The last approximation assumes that the radius of the magnetic sphere is identical to its hydrodynamic radius. It is important to realize that both velocity components grow logarithmically with twice the number of particles in the cluster. The geometric details of the slender body are then provided via the constants $C_{∥}$ and $C_{⊥}=C_{∥}+1$ in Equation (23), which are given by (Equations (7.12)–(7.14) in and (7.23) in [34])
(24)$$\begin{array}{ccc} \mathrm{cylinder}: & C_{∥}=-\frac{3}{2}+\ln2\approx -0.8069 & \mathrm{and}\;C_{⊥}=+0.1931\;,\\ \mathrm{spheroid}: & C_{∥}=-\frac{1}{2}=-0.5 & \mathrm{and}\;C_{⊥}=+0.5\;,\\ \mathrm{double}\;\mathrm{cone}: & C_{∥}=-\frac{1}{2}+\ln2\approx +0.1931 & \mathrm{and}\;C_{⊥}=+1.1931\;. \end{array}$$

Not surprisingly, the velocity parallel to the long axis v ∥= 2v⊥−κ is roughly twice as big as perpendicular to it [17]. The geometric features in Equation (24) namely cylinder and double cone (spindle) are good matches for the shapes that one finds among the particle aggregates (cf. Figure 6b–d). Such spindle-shaped clusters form at higher local concentrations after a certain length (*n* ≥ 15) of the initial line-growth of the cluster. This is schematically explained in Figure 6e. If there are sufficiently many particles in the vicinity, such clusters will form in any magnetic field and this behavior is not a particular feature of the suggested instrument. However, they can easily be studied in them macroscopically [17] and microscopically [33].

If some typical numbers are plugged in Equation (21) ($M_{s}^{m}$ = 50 Am^2^/kg, ρ = 2500 kg/m^3^, *G* = 1 T/m, η = 1 mPas) and the velocity is calculated for various *R* = *R*_h_, however, here the time, *T*, to travel 1 mm is given as: *R* = 1 nm → *T* = 1.1 a, *R* = 100 nm → *T* = 1 h, *R* = 10 μm → *T* = 0.36 min, the latter is roughly the size of an erythrocyte, which has to fit through all blood vessels. Hence, to travel a biologically relevant distance in decent time, large particles with strong magnetization should be used. However, the size of the particles used inside biological systems is often considered of paramount importance, because larger particles are more likely to be recognized as foreign bodies and will be attacked by the immune system and might cause clotting. In a way, guiding makes this point somewhat less important, because the big particles are injected somewhere and subsequently guided to their target position (like an endoscope). The only processes to worry about are those which could hamper this transport. In this picture, it also does not matter if the big particle is solid or an aggregate from a large number of smaller particles. Although the latter sounds more promising because such agglomerates would be somewhat flexible in shape and can disintegrate at the target site by removing the homogeneous field (using two dipoles of equal strength). In this way, a release process may be triggered. Technically, magnetic field gradients inside humans are limited to 1–10 T/m (cf. Figure 4b), and if all other parameters are optimized, a speed of 1 mm/s still requires particles with sizes from 100 nm to 1 μm.

All the above is again valid for ideal Newtonian liquids only. Guiding through a living organism, one will encounter many more problems, as body liquids—foremost blood—show complex rheology because they contain living deformable cells. Additionally, for guiding through blood vessels, the strong, pulsatile flow needs to be overcome. When dealing with 3D systems (see next section), gravitational and buoyancy effects need to be considered as well.

## 6. Possible 3D Designs and Imaging

So far, only instrumentation to guide paramagnetic objects across a plane was presented. However, almost all applications need to have 3D-control over the object’s trajectory. The most straightforward solution to this is to use Halbach-spheres [35] instead of cylinders (see Figure 7a). These are constructed by rotating the cross-section of a Halbach-cylinder around an axis through opposite poles. One may wonder how a sample can be inserted into a closed sphere, but there are certain angles at which such spheres, likewise cylinders [36], can be opened without a magnetic force [37]. Nonetheless, other problems will arise when using rotating nested spheres. Maybe most challenging will be the fact that the stray field of such Halbach spheres is no longer zero, so that significant torque can be expected. Alternatively, one could think about a system of rotatable cylinders on a spherical gantry to get the additional degree of freedom (see Figure 7b). This might be interesting because it also allows to decouple the force field of a quadrupole from the orientation of a dipole by simple rotation in a plane where both cylinders are orthogonal. Such a device could be a reasonable option to move macroscopic and ferromagnetic objects (e.g., catheters, capsules, or micro-robots) that possess very strong magnetization, so that the guiding fields do not have to be particularly strong. Applications with nanoparticles cannot afford the extra space required for full 3D rotation around an elongated object like a human or most animals, and the magnetic forces between such gimbals will also become significant.

In [17], it was suggested to maximize the field in the center of the machine to keep the guided object just there while moving the container in the third dimension. However, this would generate a rather metastable situation, because in the front and back of these homogeneous regions, the field drops with a strong gradient or force components out of the central plane.

Therefore, a new design is suggested here. The system with all the features described and analyzed in Appendix D is sketched in Figure 7c. Different to Appendix D, all magnets have finite length. To homogenize the dipolar field in the center, the Halbach dipoles are arranged with a gap (calculated in Appendix E) in which the quadrupole pair is inserted to compact the apparatus. If the dipole pairs point in the same direction (Figure 7c), the situation of the ideal system is maintained in a central *xy*-slice. Here the object can be moved and accelerated via rotation of the quadrupoles (as demonstrated in [17,33]). Now, if the quadrupoles are put in a cancelling position (cf. Figure 3j) and one of the dipole pairs is inverted (see Figure 7d), a gradient $\partial B_{xy}/\partial z$ is generated, which is constant in the central plane. This will then move the object in the *z*-direction with velocity *v_z_*, which can be scaled by controlling the strength of the dipole-pairs (cf. Appendix E). In a real system it would be recommendable that the container in which the object is guided is moved by approximately the same speed (e.g., on a stage movable along the axis like in CT- or MRI-scanners), but in opposite direction, to keep the object in the center of the machine.

The instrument in Figure 7c,d then combines 3 degrees of freedom in guiding with velocities *v_x_*, *v_y_* (via rotating the quadrupoles, cf. Figure 3h–j,l and Figure 7c), and *v_z_* (by bringing the dipole pairs in opposite direction, cf. Figure 7d) with 2 degrees of freedom in orientation in *x* and *y*, e.g., of nanoparticle chains by co-rotating the dipole pairs (cf. Figure 3c–e,k). Additionally, all these degrees of freedom can be scaled by the apparatus sketched in Figure 7c,d. Orientation along *z* is not possible because ideally there is no such field component in the center of the machine.

A crucial aspect of guiding is the control of the actual position of the steered object. If the container in which the object is moved is not transparent, like most biological systems, a blindfolded tour will probably not end up at the designated target position and maybe even cause severe inner injuries. Maybe with an exception for some endoscopic or capsules devices with an onboard camera, the position must be continuously monitored by a non-invasive imaging technique. Since the proposed magnet system already possesses strong homogeneous and gradient fields, there are two obvious candidates for this: magnetic resonance imaging (MRI) and magnetic particle imaging (MPI).

*Magnetic Resonance Imaging* (for excellent introductions see [38,39]) uses the magnetic moments of atomic nuclei (foremost ^1^H), which are associated with the spin of the nucleons. To generate an energetic difference between the different orientations of the spin, a very strong and homogeneous (polarizing) magnetic field is needed. Typically, the energy differences between these energy levels correspond to a radiofrequency (e.g., ca. 42 MHz/T for ^1^H). To excite transitions between the energy levels, a magnetic field must be irradiated with a frequency that matches the strength of *B*_0_ and is oriented perpendicular to it. This is usually done in the form of short bursts of an AC current of matched frequency in a coil that surrounds the sample. The nuclear spins then induce a much weaker signal in the same coil. The signal strength scales with $B_{0}^{2}$ and therefore magnets must be strong for this technique. Spatial resolution is then introduced by constant gradient fields, *G*, i.e., a linear variation of the magnetic field component, which is parallel to *B*_0_ along the spatial direction of interest. As said before, the same requirement (*B*_0_ > *Gr*) as in Equation (7) is necessary to create bijective spatial encoding. Because the frequency of the MRI-signal is directly proportional to the magnetic field, such a gradient generates an integral projection of the sample along the selected gradient direction. Therefore, the instruments presented here possess all requirements for MRI, albeit the gradients discussed so far would be way too strong for MRI (*G* < 0.01 T/m) and two quadrupoles would be needed to allow for guiding at high *G* and imaging at low *G*, also ensuring the objects are no longer moving while being imaged. Since the gradients in the discussed designs can be rotated, projections at various angles can be easily acquired and reconstructed to Cartesian coordinates using an inverse Radon transform, aka “back-projection” as used in CT-scanners [40]. The only extra-equipment would be an rf-coil with an amplifier and the MRI-spectrometer. Because the contrast of MRI in biological systems mainly stems from the magnetic interactions of the nuclear spins in water with the direct environment, superparamagnetic nanoparticles cause a strong disturbing effect and are already standard MRI contrast agents [38,41]. However, this leads the reader to believe that MRI comes for free with the suggested guiding systems. The difficulty lies in the field homogeneity required for MRI. While a guiding system could sufficiently function with a precision of 1%, MRI requires fields that are 2–3 orders of magnitude more homogeneous. Nevertheless, recently great progress was made in building functioning MRI systems using Halbach-magnets [42,43].

*Magnetic Particle Imaging* [44,45] relies on the presence of superparamagnetic particles to generate a signal. Their magnetization follows a magnetic drive field oscillating at a suitable frequency (typically in the kHz–MHz range depending on the particle relaxation behavior) and its time derivative induces a signal in a receive coil. It can be distinguished from the drive field because the magnetization in such materials does not depend linearly on the applied field (typically *M*(*H*) is described by a Langevin-function) and can be identified (e.g., via overtones in the spectrum) or filtered from the induced voltage. This signal is then proportional to the concentration of the particles. Spatial resolution is obtained by applying strong gradient fields, so that the particles magnetization outside the region of zero-crossing is saturated, hence it cannot follow the drive signal and no longer gives a signal. By moving such zero-lines or points through the region of interest, images of the particle concentration can be reconstructed. The latter are typically those magnetic fields that consume most of the energy in electromagnetic MPI-devices, but can be generated with the described apparatus by combining quadrupole and dipole fields as suggested in [46].

Apparently as the title of this section suggests, none of the discussed devices have been built yet. Of course, other tracking possibilities exist as well, especially for larger objects as reviewed in [47].

## 7. Applications

So far two types of guiding systems were designed, larger ones with *R*_i_ ≈ 5 cm (the original MagGuider from [17] and a system with one dipole and two compensated quadrupoles, i.e., system M1 in [33], cf. Figure 8a,b) and several smaller ones with *R*_i_ ≈ 1.5 cm made to fit light-microscopes (for instance M2, M3 in [33], cf. Figure 8c,d). Appendix G contains some practical suggestions for building such magnet systems.

Several objects were guided in 2D with such systems. Being equipped with two quadrupoles of almost equal strength allows to accelerate the object or even to bring it to a stop. To illustrate this option, a 1-mm steel ball was used to write the initials of the author’s employer (cf. Figure 9a or [48]). The steel ball was chosen because it can easily be tracked visually or by a regular camera, however, in order to slow down its movements in the very strong gradients of the device shown in Figure 8b it had to be immersed in an extremely viscous silicon oil. On the other hand, the cobalt ferrite nanoparticles shown in Figure 9b,c had almost 10 times the velocity of that steel ball; however they were immersed in water (with a 50,000 times lower viscosity than the silicon oil used in Figure 9a). They are only visible (average diameter of 75 nm) because they form very long clusters (*n* = 10^5^–10^6^). Several other commercial superparamagnetic nanoparticles were compared under identical conditions in [17].

In [33], it was shown that not only bare nanoparticles can be moved but also human and murine cells that have taken up commercial nanoparticles (cf. Figure 9d,e). This might be important for ex-vivo 3D-printing of tissues or entire organs. In-vivo regenerative medicine might make use of this (e.g., lesion of nerves, retina, inner ear, etc.) by transporting specific cells to lesions, building cellular scaffolds, and repairing or engineering tissues with certain textures.

The last example is like a micro-robot including actuation, however, on a much smaller scale. In [49], liquid crystalline elastomers, which included iron-oxide particles, were functionalized with thermo- and photoresponsive groups. The long distance transport could then be provided by the suggested magnetic guiding system. At the target site, material properties could then be changed either by heat or light and used for actuation. For instance, the stickiness of the particles was modified, so they could adhere to some other non-magnetic object, and both could then be moved together magnetically. Bringing the liquid crystalline elastomer to the isotropic phase by heating weakened the adhesion and released the object (cf. Figure 9f).

## 8. Conclusions and Outlook

Magnetic guidance of non-charged objects by combining a strong homogeneous magnetic field with smaller deflection fields is a very general concept that can be put into practice in various ways using resistive or superconducting electromagnets, by permanent magnets or hybrid systems. This article focusses on permanent magnets and in particular in Halbach configuration, because they are inexpensive, stronger, and can directly be scaled up. Of course, there are alternative technical routes to fulfill this task than using nested dipolar and quadrupolar Halbach cylinders in the presented combinations. Alternative arrangements might be more useful and compact for certain applications. For instance, in [50], six permanent magnets are individually rotated to generate the desired fields. This article limited the discussion to Halbach magnets because the author believes that they produce magnetic fields closest to the demanded ideals and are easy to calculate and understand.

If orientation and force velocity do not have to be controlled independently, dipole and quadrupole can also be combined in a single ring and so forth. In the end, the number of degrees of freedom needed for a particular experiment decides upon the complexity of the movable rings of the magnetic device.

The techniques proposed here cannot only be used to guide superparamagnetic nanoparticles. This particular application is probably the most demanding in terms of required forces and hence gradient and field strengths. Much weaker fields would be sufficient to guide objects that contain larger ferromagnetic parts (e.g., endoscopes, mini-robots). The necessary magnet systems for this application could therefore be much more spacious than the devices shown in Section 7. It also would not be a problem to power actuation of such robots by additional rf fields. Nevertheless, guiding superparamagnetic nanoparticles is still the grand challenge in this field, and a few routes were shown how this could be accomplished, e.g., by designing larger clusters of nanoparticles. For instance, a “pill” made from differently functionalized particles could be synthesized. It could contain some drug-carriers placed in the core and covered with layers of biocompatible material, all superparamagnetic and weakly cross-linked, such that it decomposes after a certain time at the target site and deploys its therapeutic payload for local treatment. In order to get a step ahead in this direction, Section 5 and Figure 4 provide some clues on the physical limits with today’s technology, and what sizes and shapes are needed to make real tools.

Taken together, the author thinks that the proposed guiding system may serve as a general solution to the problem of steering by magnetic fields. It is hoped that this mainly conceptual overview convinces researchers that this methodology has the potential to simplify magnetic guiding, make it more versatile while reducing its costs and allowing for human scale applications. To achieve this, much more efforts are needed in designing proper computer-controlled systems with sufficiently homogeneous fields to implement all the different modalities that were suggested in this article.

## 9. Patent

The principle of the presented guiding with permanent magnets is patented in O. Baun, and P. Blümler “Vorrichtung zur Bewegung von magnetischen Partikeln in einem Raum mittels magnetischer Kräfte” DE102016014192A1, 2016.

## Figures and Tables

**Figure 1 cells-10-02708-f001:**
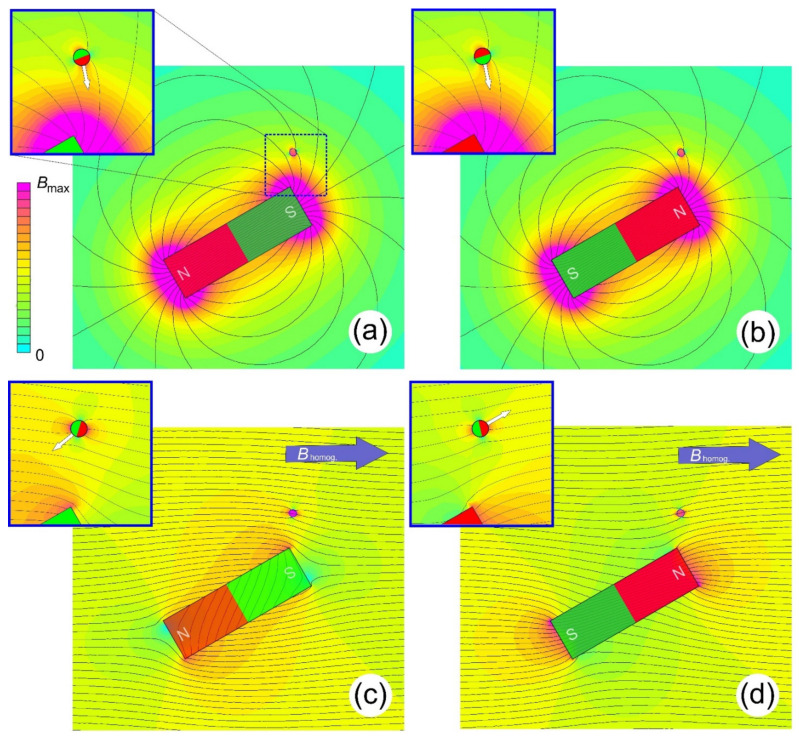
Illustration of the suggested guiding principle. A small magnetizable sphere serves as the object to be guided by a large deflecting bar magnet. The colors indicate the magnitude of the local magnetic flux density (see color bars on the left, *B*_max_ in (**a**,**b**) is roughly a quarter of that in (**c**,**d**)). The black lines are field lines. A zoom of the region around the object is shown on the inserts. The top rows (**a**,**b**) just show the field generated by the deflection magnet, while in (**c**,**d**) a strong and homogeneous field is superimposed to the scenario above. The difference between the columns is the orientation (south- and north-pole) of the deflection magnet. (**a**,**b**) Changing the magnet’s orientation has no effect on the movement of the object (white arrow), because the object is magnetized in opposite directions as well and just moves to the highest flux density. The additional homogeneous field in (**c**,**d**) essentially keeps the magnetization direction of the object along its horizontal direction. The field of the deflecting magnet now causes the opposite magnetic “landscape” around the object and hence moves in opposite directions. The data were generated using FEMM (www.femm.info) but should serve for illustration purposes only.

**Figure 2 cells-10-02708-f002:**
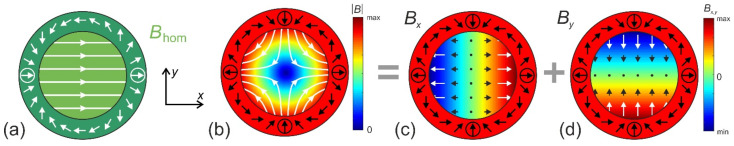
Sketch of ideal Halbach cylinders: (**a**) inner dipole with a homogeneous field of strength, *B*_hom_ along the *x*-axis. (**b**) Inner quadrupole with a circular modulus field, which can be decomposed into two linear field components *B_x_* = *Gx* in (**c**) and *B_y_* = −*Gy* in (**d**). The hollow cylinders consist of permanent magnet material with continuously changing magnetization direction (arrows). The poles are encircled. In (a,b), the magnetic field is represented by field lines, while in (**c**,**d**), the arrows are field vectors (the different colors are only for better contrast). Note that the magnetic fields are only inside the hollow cylinders and that there are no stray fields.

**Figure 3 cells-10-02708-f003:**
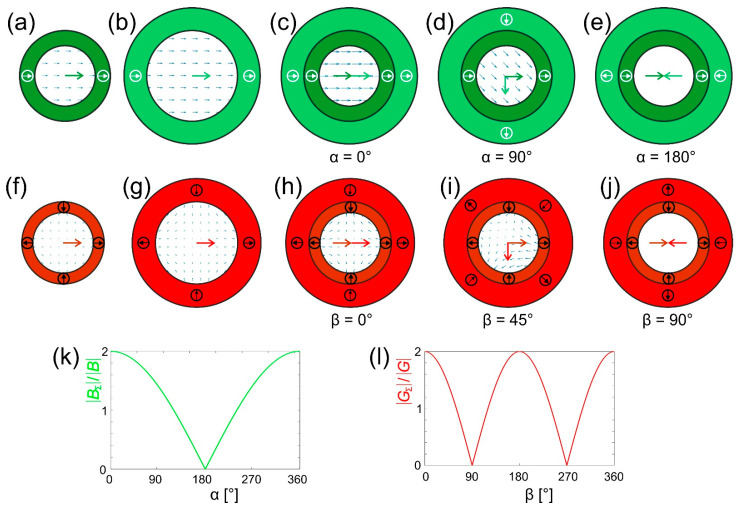
Coaxial arrangement and rotation of Halbach dipoles (green, upper row) and quadrupoles (red, lower row): (**a**,**b**) two Halbach dipoles that produce the same field strength, $\vec{B}$ (central green arrow); are coaxially nested in (**c**–**e**) and the outer one is rotated by an angle α. The resulting field is also illustrated by $\vec{B}$-arrows. (**c**) For α = 0°, the fields are parallel and the two dipole fields add to 2$\vec{B}$. (**d**) For α = 90°, the fields are orthogonal and the two dipole field vectors add to $\sqrt{2}\left|\vec{B}\right|$ at an angle of 45°. (**e**) For α = 180°, the fields are antiparallel and cancel each other. An analog presentation is shown in (**f**,**g**) for two nested quadrupoles that produce the same field gradient (i.e., the derivative of the field. The red arrow shows the horizontal component only). (**h**–**j**) Same representation as above. Note that the gradient rotates at twice the angle of the quadrupole (cf. Appendix C). (**k**) The angular dependence of the combined field of both dipoles, $B_{\Sigma }\;=2\left|\vec{B}\right|\left|\cos(\alpha /2)\right|$; (**l**) angular dependence of the gradient strength of the two quadrupoles $G_{\Sigma }\;=2\left|\vec{G}\right|\left|\cos\alpha \right|$.

**Figure 4 cells-10-02708-f004:**
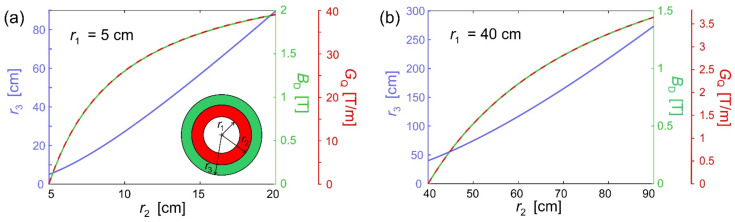
Estimation of the achievable magnetic fields and gradient strengths for a simple guiding system consisting of an inner Halbach-quadrupole (red) and an outer Halbach-dipole (green) (see sketch of the geometry inserted in (**a**)). Shown is the dependence of the homogeneous field, *B*_D_ (green line), produced by the dipole and the gradient, *G*_Q_ (dashed red line), of the quadrupole versus the inner radius of the dipole *r*_2_. Both have the same functional dependence but scale with *r*_1_. Therefore, there are two axes on the left in the respective colors. In (**a**), the internal radius is set to *r*_1_ = 5 cm, which could allow for guiding objects inside a rodent. The innermost radius in (**b**) of *r*_1_ = 40 cm was chosen to host a human. A remanence *B*_R_ = 1.3 T was used in both cases. The outermost radius, *r*_3_, *B*_D_, and *G*_Q_ are then given by Equation (15).

**Figure 5 cells-10-02708-f005:**
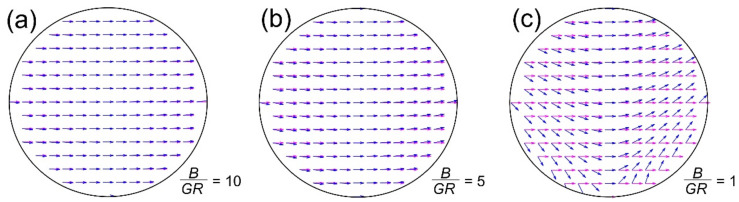
Schematic representation of the effect of concomitant gradients or violating Equation (7). Shown is the force over the internal area with a radius *R* for different ratios *B*/(*GR*), cf. Equation (7). The magenta-colored arrows ignore the local field and $\vec{m}$ always points along $\vec{B}$ (as in Equation (6), while the blue arrows account for the full field. This shows the increasing deviation from a homogeneous force direction with increasing gradient and/or distance from the center. (**a**) *B*/(*GR*) = 10 or δ_max_ = 5.7°, (**b**) *B*/(*GR*) = 5 or δ_max_ = 11.5°, and (**c**) *B*/(*GR*) = 1 or δ_max_ = 90°. See text for more details.

**Figure 6 cells-10-02708-f006:**
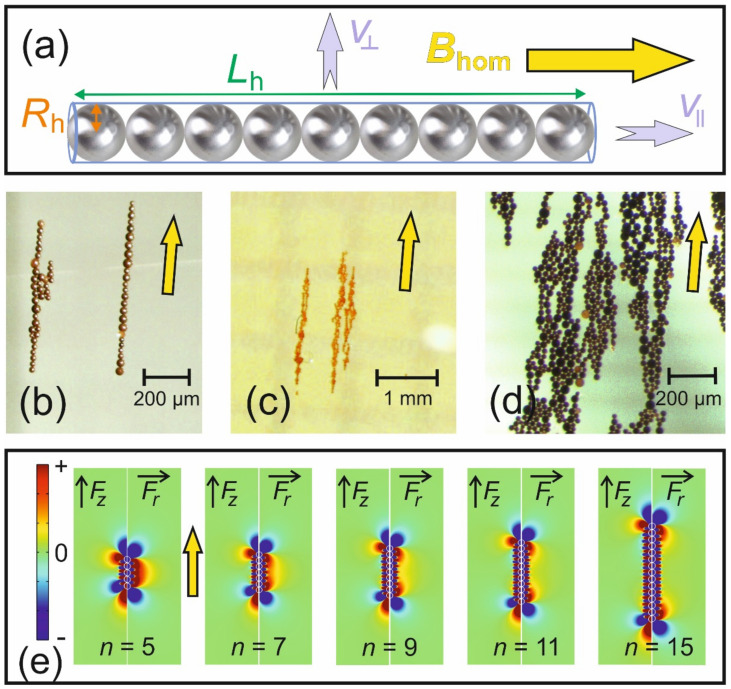
Behavior of superparamagnetic particles under magnetic guiding conditions: (**a**) Magnetized by the homogeneous magnetic field, ${\vec{B}}_{\mathrm{hom}}$, the particles form long chains in the same direction. Ideally they are mono-disperse spherical particles with a hydrodynamic radius, *R*_h_, and then *n* of them form a chain of length *L*_h_ = *n* 2*R*_h_. However, if they are moved, the velocity of the chain has two extremes of being moved either with $v_{∥}$ along the long axis or direction of the external field or perpendicular to this direction with $v_{⊥}$. (**b**–**d**) Shows microscopic photographs of typical clusters of magnetite particles with an average diameter of 30 μm. The yellow arrow indicates the direction of ${\vec{B}}_{\mathrm{hom}}$. (**b**) At low local concentration linear beaded chains (right) are typically formed. After a certain length particles can also attach from the sides, forming spindle-shaped structures (left). (**c**) At higher concentrations such spindles dominate and separate from each other. (**d**) Such “carpet-like” structures are also observed at very high concentrations and at fluid surfaces. (**e**) FEM simulation (COMSOL 5.5) showing the magnetic force (Equation (1)) in cylindrical coordinates (*r*, *z*) around chains of *n* particles, which are all magnetized in the vertical (+*z*) direction. For a small number of particles; the “repulsive“ part (red) of the force dominates in radial direction. The particles must first overcome this force, to enter the attractive force minima (blue) closer to the chain. This repulsive force vanishes at the center for *n* ≥ 15. Therefore, particles aggregate at the sides of longer chains in a hexagonal pattern, causing the typical spindle-shapes.

**Figure 7 cells-10-02708-f007:**
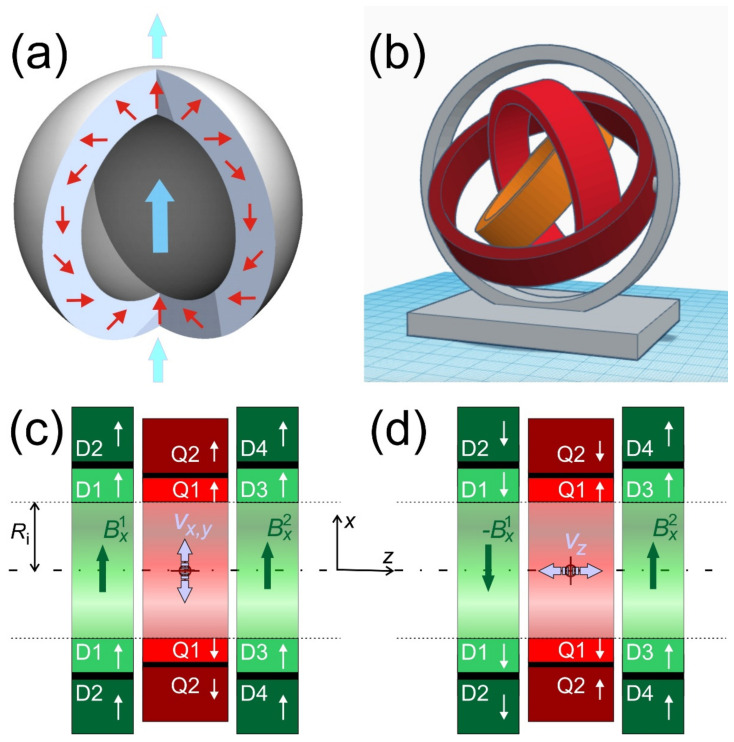
Three-dimensional guiding ideas: (**a**) A Halbach sphere is made by rotating a Halbach cylinder around the axis through the poles. The magnetic field inside is 4/3 higher than in an ideal Halbach cylinder, but it also produces stray fields (small vertical arrows). (**b**) Sketch of 3D gimbal gantry. (**c**) Axial cross-section through a magnet system of two pairs of dipoles (green) and two quadrupoles (red). The dipoles and quadrupoles could for instance be concentric Halbach cylinders, here displayed as axial slices. The dipoles are split in a pair to homogenize the field in the center. If the strength of D1 = D3 and D2 = D4 the system is described in Section 3 and analyzed in Appendix D. In the shown configuration, objects are oriented in the *x*-direction and moved in the central *xy*-plane (lilac arrows). (**d**) Adding one additional degree of freedom to the system in (**c**) by independently rotating the left and right dipoles. If the quadrupoles are put in a cancelling position and the dipoles are in opposite directions, a gradient along the *z*-direction (axis) is generated. This system is discussed in Appendix E. If the guided object moves, then with *v_z_* the container (e.g., patient) should be moved via a stage with −*v_z_* so that the object stays in the center.

**Figure 8 cells-10-02708-f008:**
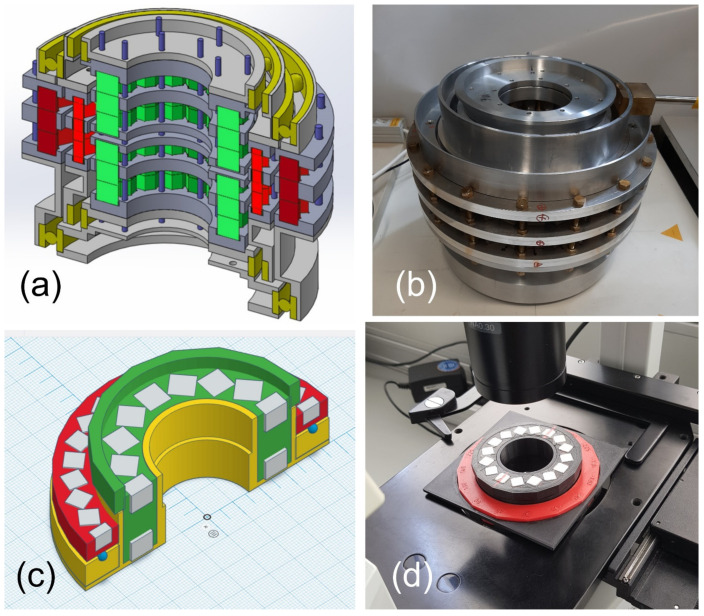
Examples of two constructed guiding systems: (**a**) Cut-away view of a schematic drawing of the larger device. The innermost opening has a diameter of 10 cm, outer diameter is 36 cm, and the total height is 27.5 cm. It consists of a static Halbach dipole (magnets in green) and two quadrupoles (darker and lighter red) that can be rotated. In gray are aluminum and in blue brass supports; ball bearings are in yellow. (**b**) Constructed system with a homogeneous field of 325 mT and gradients from 0 to 2.1 T/m. The complete system has a weight of ca. 100 kg. (**c**) Cut-away view of a schematic drawing of the smaller device; the larger squares on the light-blue pad have a size of 1 cm. The magnets in both dipole (green with two magnet layers) and quadrupole (red) are shown in gray. Supports are in yellow and glass balls, which serve as bearings, are in blue. (**d**) 3D printed version of (**c**) used in a light microscope. It has a homogeneous field of ca. 100 mT and a gradient of ca. 1.3 T/m, which both can be rotated independently. It weighs 273 g. More information about both systems can be found in the supporting material of [33].

**Figure 9 cells-10-02708-f009:**
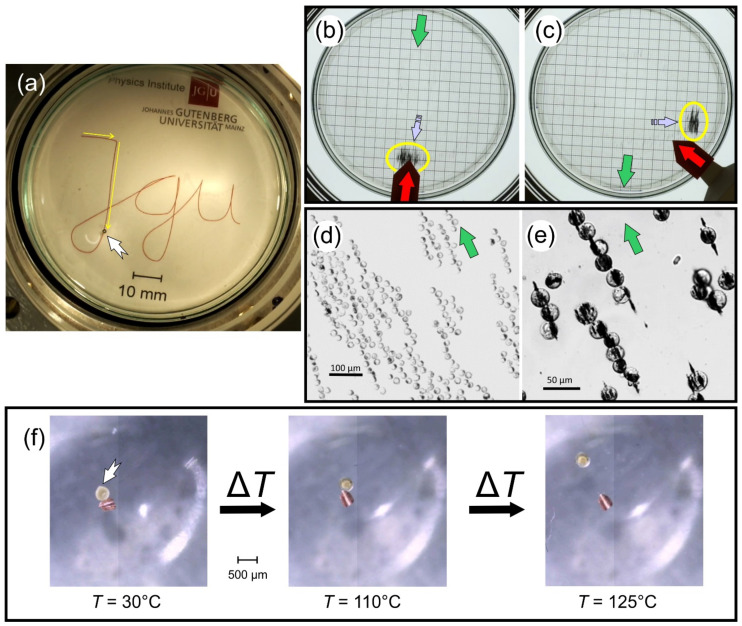
Various applications of the presented guiding systems. (**a**) A 1-mm steel ball (white arrow) was used to write the letters JGU using the magnet system presented in Figure 8a,b. The yellow arrows indicate the initial motion. (see also Supplementary Video S1 in [33,48]). To slow down the speed of the ball it was immersed in very viscous silicon oil (η = 50 Pas). Nevertheless, it had an average speed of 1.45 mm/s. (**b**,**c**) A small cloud (yellow ellipse) of cobalt ferrite nanoparticles with an average diameter of 75 nm was moved in water with a magnet system consisting of one dipole and one quadrupole. The direction of the static *B*_hom_ = 0.1 T is indicated by a green arrow while the quadrupole with *G* = 0.2 T/m can be rotated and is marked by a red arrow. The resulting force or velocity (ca. 14 mm/s) is marked by a lilac arrow. The underlying squares have a side length of 5 mm. (**c**) Image taken 3.5 s after the quadrupole (red arrow) was moved by ca. 45° (Supplementary Video S2 in [17]). (**d**) Murine macrophage cells incubated with 100 nm iron oxide particles were guided in a system like in Figure 8c,d while being studied with a light microscope. Due to the strong and homogeneous dipole field, they form long clusters that are reversible when the magnetic field is removed. (**e**) Close up of (**d**) the cells arranged around the connected nanoparticles like meat pieces on a shish kebab (for details see [33]). (**f**) A liquid crystalline elastomer with incorporated iron oxide particles (white arrow) is guided over long distances (not shown) with a system like in Figure 8c,d. When the particles arrive at the target site, actuation can be initialized by a temperature change. A small piece of copper sticks to it at temperatures up to 110 °C (left & middle image) but leaves on the right when 125 °C are exceeded (Supplementary Video S7 of [49]).

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
