# Peer review of "Magnetic Guiding with Permanent Magnets: Concept, Realization and Applications to Nanoparticles and Cells"

_cells, 2021, doi:10.3390/cells10102708_

Round 1

Reviewer 1 Report

This manuscript contains a description of the remote magnetic guiding by permanent magnets. The Author discusses the special hydrodynamics of nanoparticle dispersion, the possibility of 3D guiding and magnetic imaging techniques. Besides, the results in guiding macroscopic objects were also presented. Of particular interest are the results of the behavior of superparamagnetic particles under magnetic guiding conditions and examples and applications of the presented guiding systems. The article certainly deserves publication in the journal “Cells”. Minor notes are below:

1) The introduction uses a large number of phrases related rather to the scientific journalistic than to the scientific language (Ex: «At the beginning of each scientific manuscript it is always good practice to define…», «It should be understood as a remote…», «Broadly spoken…», etc.). In itself, this is not something unacceptable, but at the same time it is not recommended for use in scientific journals;

2) It is good that the author marks some important points from the point of view of terminology at the end of the introduction, but it should be more specific to indicate the results achieved and to cover the obtained experimental systems more broadly;

3) The author has described part 2 (Concept of magnetic guiding) in too much detail. Some well-known equations could be omitted (along with the scripture), leaving a few relevant references;

4) In the text and the description of Figure 6, there is a controversial formulation «small superparamagnetic particles» since superparamagnetic particles are in the nanometer range by default, so this clarification is unnecessary.

Reviewer 2 Report

The paper presents an extensive review of magnetic guiding using permanent magnets, particularly focused in Halbach systems. The concept and realization are very well discussed from the equations of magnetic fields and forces to the velocities and physical limits. The applications to nanoparticles and cells are quite limited maybe due to the limited number of reports in the literature. The field of this review is of high relevance and the scientific quality of the manuscript is high. I consider that this work should be published in Cells after some minor considerations:

  • In general the paper is very well written, but maybe some kind of sentences can be avoided: pag 2, line 33: “the author wants to apologize for using a 33 somewhat sloppy terminology concerning magnetic flux density” or “r, but also as allusion to pop culture because it was quickly assembled in a day 45 from scrap and left overs…”.
  • In Fig. 1 and 2 the author should explain the program used to obtain the images. Did you used a software to make these magnetic simulations, or they are only illustrative schemes?
  • What is the magnetic material (the composition of the magnet) used to obtain the results plotted in figure 4?
  • Figure 6 should appear in page 11 or 12, not 2 pages after (the same happens with fig 7).
  • For magnetic guidance in in vivo systems (human or animal models) the blood velocity should be considered. Can the author introduce this variable in the discussion?
  • The schemes of Fig 7 c) and d) are not very illustrative….maybe a 3D scheme (like the used in 7 a) and b)) can help.
